# Dietary Patterns in Adolescent Obesity as Predictors of Long-Term Success Following an Intensive Inpatient Lifestyle Programme

**DOI:** 10.3390/ijerph192416613

**Published:** 2022-12-10

**Authors:** Jana Brauchmann, Anne-Madeleine Bau, Gert B. M. Mensink, Almut Richter, Andrea Ernert, Theresa Keller, Susanna Wiegand

**Affiliations:** 1Center for Chronically Sick Children, Charité Universitätsmedizin Berlin, 13353 Berlin, Germany; 2Robert Koch Institute, 13353 Berlin, Germany; 3Institute of Biostatistics and Clinical Epidemiology, Charité Universitätsmedizin Berlin, 10117 Berlin, Germany

**Keywords:** childhood obesity, pediatric obesity, dietary pattern, weight reduction, inpatient rehabilitation

## Abstract

(1) Background: Lifestyle interventions for adolescents with obesity show minor long-term effects on anthropometric parameters. The persistence of dietary changes after obesity inpatient rehabilitation has not been sufficiently investigated. (2) Objectives: To analyse dietary patterns in German adolescents with obesity as predictors of long-term success following an intensive inpatient lifestyle programme regarding food choices as well as body weight and comorbidities. (3) Methods: Food consumption data of 137 German adolescents with obesity aged 10-17 years were collected by a nutrition interview. Cluster analysis was used to group the participants according to their food consumption. Dietary patterns, changes in body weight and insulin resistance were compared over a 2-year-period. (4) Results: Three dietary patterns were identified. Big Eaters (n = 32) consume high amounts of total sugar and meat, Moderate Eaters (n = 66) have a diet comparable to the national average, and Snackers (n = 39) have a particularly high consumption of total sugar. Big Eaters and Snackers significantly reduced the consumption of total sugar. Among Moderate Eaters, no persistent changes were observed. (5) Conclusion: Weight reduction interventions can induce long-lasting changes in the diet of adolescents with obesity. Therefore, the success of a weight reduction intervention should not be determined by weight reduction only.

## 1. Introduction

Globally, the prevalence of adolescent obesity increased between 1975 and 2016, and has stabilized at a high level in many high-income countries since 2000 [1]. In Germany, obesity affects 6.5% of girls and 8% of boys in the 11- to 13-year-old age group. Among 14- to 17-year-olds, the rates are even higher: 7.7% of girls and 9.2% of boys [2]. Already in adolescence, obesity increases the risk of comorbidities, including hypertension, dyslipidemia, hyperglycemia and nonalcoholic fatty liver disease [3] as well as leading to psychosocial problems such as low self-esteem and the risk of body dissatisfaction [4]. Obesity in the early years of life can result in considerable costs for the health care system and due to productivity losses [5] because most of the affected young individuals continue to live with obesity throughout adulthood as few succeed in consistent weight reduction [6]. Due to the comorbidities and the need for long-term medical care, obesity can be classified as a chronic disease [7]. The successful treatment of obesity in adolescence is therefore an important issue.

Dietary modification is an essential aspect for weight reduction [8]. Therefore, an increased understanding of food consumption in adolescents with obesity is a necessity for effective weight loss therapy of this group. A commonly used method to examine food consumption in a defined population is the dietary pattern analysis. Instead of focusing on isolated foods and nutrients, the analysis of dietary patterns can provide comprehensive insights on the impact of diet on health as it better reflects the complex interplay of diverse food intake [9].

Previous research has focused on either dietary patterns of the general adolescent population or adults with obesity. In regard to the general adolescent population, dietary patterns with a high consumption of sweets and chocolate, refined grains, processed meat, and low-fiber foods increase the risk of developing overweight or obesity [10]. Similarly, high consumption of sugar-sweetened beverages is an important dietary determinant of obesity in adolescents [11,12]. It is also known that dietary patterns with a high consumption of ultra-processed foods that are rich in fat, salt, and refined carbohydrates are also associated with a higher risk of cardio metabolic alterations in adolescents [13]. The terms “western” or “westernised”, “fast food”, “modern”, “high fat and sugar” are used for dietary patterns determined to be “unhealthy” in this context [14]. Most studies in which dietary patterns are described are cross-sectional studies and use a semi-quantitative food frequency questionnaire to collect food intake data [14].

In adults with extreme obesity undergoing bariatric surgery, preoperative dietary patterns called “big eater”, “snackers”, “sweet eater”, and “light-soda consumer” have been described [15].

Due to a lack of research on the dietary patterns of adolescents with obesity, there are still open questions that need to be addressed. Can unique dietary patterns be identified in adolescents with obesity? Are dietary patterns influenced by an inpatient weight loss intervention? Do adolescents with a certain dietary pattern benefit more than others with a different dietary pattern in terms of weight reduction in the long-term?

To answer these questions, the presented research has two main objectives. The first aim is to identify baseline dietary patterns in adolescents with obesity. The second aim is to investigate the long-term effect of a weight loss intervention on food selection and weight progression depending on the baseline dietary patterns. This research will focus on, and therefore be limited to, German adolescents with obesity.

## 2. Materials and Methods

Study population and baseline characteristics. The analyzed nutritional data were obtained from the pediatric clinical trial of the MAINTAIN study (Clinical Trials NCT00850629), also referred to as “ped-Z-Project” 218/1. The study was a randomized controlled trial (RCT). The RCT took place from October 2009 to 2015 and is described in detail elsewhere [16].

There were 180 adolescents with obesity recruited for the study. Of these, 147 adolescents and their families met the following inclusion criteria: Age between 10 to 17 years, BMI above the 97th percentile at recruitment, active participation in the inpatient rehabilitation program as well as the intervention and control group as well as the free living part [16]. Anthropometric data of the participants were measured in the following procedures. Body weight was measured while participants were wearing light underwear, without shoes on a digital scale (Soehnle, Backnang, Germany) with an accuracy of 0.1 kg. Body height was measured according to the international recommendation with the head oriented in the Frankfurt horizontal plane [17]. The body height was measured with a wall-mounted stadiometer (Keller, Germany) having an accuracy of 0.1 cm. The Body Mass Index (BMI) was calculated using the quotient of body weight (kg) and body height (m^2^). To determine the degree of overweight, age- and sex-specific changes in BMI must be taken into account in children and adolescents according to the guideline of the German Obesity Society [18]. Therefore, the standard deviation score of the Body Mass Index (BMI z-score) was calculated and compared with age- and sex-specific German reference data published in 2001 [19]. Obesity was defined as having a BMI above the 97th percentile [18]. The different stages of puberty among the participants were determined according to the Tanner stages [20,21] after medical examinations. The participants were categorized as infantile (Tanner Stage 1), pubertal (Tanner stages 2–3) or post-pubertal (Tanner stages 4–6).

Study enrollment process. The flow chart presented in Figure 1 is based on the MAINTAIN study. The chart shows points in time within the study when the nutritional data was collected.

A number of 180 adolescents and their families were enrolled in the study. A total of 147 of the 180 fulfilled the inclusion requirements and received a foundational consultation for a healthy lifestyle according to the “BABELUGA lifestyle-monitoring map” [22]. The consultation took place at the Pediatric Outpatient Obesity Clinics of the Charité Universitätsmedizin Berlin. The nutritional advice was based on recommendations from the German food based dietary guidelines for adolescents, referred to as the Optimized Mixed Diet (OMD) [23]. The OMD corresponds to a well-balanced diet adhering to the energy needs and nutritional requirements of adolescents. It is used as a standard of quality for school meals in Germany and is recommended for an optimal diet for children and adolescents with a BMI z-score over the 90th percentile. The OMD is comparable to the Traffic Light Diet from the USA [24]. 

Weight reduction phase. The 147 study participants took part in an inpatient rehabilitation program at a specialized residential weight loss center in Beelitz, Germany. The average stay in the weight loss center was 6.2+/−1.5 SD weeks. Weekly group consultation sessions with a dietician assistant were offered during the inpatient rehabilitation program, as well as the preparation of healthy meals under the supervision of a professional cook. In addition, seminars for the study participants and their parents were held every Saturday. The catering during rehabilitation was also based on the OMD recommendations. Three main meals were offered daily, consisting of a cooked lunch and one bread meal for breakfast as well as for dinner. Between the main meals, two fruit and one tea snack were offered [23]. Depending on age and sex, the study participants consumed between 1200 and 1800 kcal daily, resulting in a negative energy balance of about 500 kcal per day [23]. According to the recommendations within OMD, the intake of free sugar was limited: no sugar-sweetened beverages were allowed, sweets were offered at most three times per week in a defined limited amount. In addition to a well-balanced diet, the following were other main aspects of the inpatient rehabilitation program: pedagogical measures, psychotherapy, physical treatment and endurance training and medical care. A detailed description of the individual contents can be found elsewhere [16]. During the inpatient rehabilitation program, 10 participants dropped out of the study due to various reasons.

Randomized controlled trial and Follow-up. The remaining 137 study participants were subsequently randomized to either a control group (n = 72) or an intervention group (n = 65). Within the scope of the RCT, it was to be investigated if the intervention affects the weight progression after the weight reduction phase. For the present analyses, the division into control and intervention groups was less important because dietary patterns were already determined at the time of recruitment (T-3). The intervention group took part in a total of 10 intervention modules within 12 months. The intervention modules are described in detail in the study design paper [16]. All modules took place in the Pediatric Outpatient Obesity Clinic of the Charité Universitätsmedizin Berlin and were planned and led by a team of nutrition scientists/dieticians, psychotherapists, and physiotherapists. The control group received only usual medical care in the outpatient obesity clinic. The follow-up took place 12 and 24 months after the weight reduction phase.

Collection of nutritional data. The Software DISHES (Dietary Interview Software for Health Examination Studies, Version 2.0, Robert Koch Institute, Berlin, Germany/dato Denkwerkzeuge, Wien, Austria) was used within the MAINTAIN study to collect the nutritional data. DISHES was developed by the Robert Koch Institute [25]. DISHES enables to assess the individual food consumption in average with a reference period of the previous four weeks from the participants within a computer-based diet history interview. DISHES has been used several times in population-based studies in Germany in both adults and adolescents [25,26]. DISHES was validated for adults [27] and compared to FFQ results among adolescents [28].

Following the typical meal pattern of a day, i.e., breakfast, lunch, dinner and in-between meals, commonly consumed foods were chosen from a checklist. Similar foods not directly shown in the checklist, could be added with an integrated food database. Therefore, the interview was standardized but was still an open comprehensive assessment method. Subsequently, both the frequency of food consumption as well as the portion sizes were requested. To estimate the portion sizes, tableware models (cups, glasses, spoons, plates and bowls) were used, along with a food picture book, adapted from the EPIC study [29]. A single DISHES-Interview lasted around 45 to 60 min.

The German Nutrient Database (BLS Version 3.0, Max Rubner Institute, Karlsruhe, Germany) was used as part of the DISHES software to evaluate the nutritional values of the food. The BLS was developed as a standard instrument for the evaluation of epidemiological nutrition studies and food consumption surveys in Germany in order to guarantee the comparability of the study results. The basis for the BLS are data from technical literature, companies from the food industry and international nutritional tables [30]. The BLS is regularly maintained and updated by the Max Rubner Institute in Karlsruhe, Germany.

As shown in Figure 1, food consumption data were collected during the first 27 months of the study: at time of enrollment (T-3), directly after the inpatient rehabilitation program (T0), as well as 12 (T12) and 24 months (T24) after the inpatient rehabilitation program. At T-3 the food consumption data from the DISHES interviews was available for 137 of the 147 study participants. Two participants of the remaining 137 were unable to reduce their age and sex adjusted BMI z-score by the necessary 0.2 units during the inpatient rehabilitation program and were therefore excluded from the study. Thus, at the time of T0, data from 135 DISHES interviews was available for the presented analysis. After the RCT, ten study participants lost interest and dropped out. Nutritional data from the remaining 127 participants was collected. Until the follow-up at time T24, another 27 study participants lost interest and dropped out. Of the remaining 100 study participants, 79 DISHES interviews were conducted. Only 67 of the 79 DISHES interviews could be considered at T24 because 12 of the 79 participants had attended inpatient rehabilitation twice since time point T12.

Processing of study data. After the conclusion of the MAINTAIN study, the data was imported into a SPSS data template for statistical analysis. Nutritional variables with extreme low or high food intake values that exceeded 10 times the expected value, e.g., 2000 g broccoli instead of 200 g broccoli, were cross-checked with the DISHES entries to ensure that the correct SI-unit was used in the initial data entry from the diet history interviews of the study. The values were manually corrected when necessary.

In order to reduce the number of input variables for the cluster analyses, single foods were combined into the following food groups according to the OMD: vegetables, fruits, potatoes/noodles/rice and other grains such as couscous and millet, bread and cereals (also flakes), milk/dairy products, meat/-products, fish, butter/margarine/oil and drinking water. The intake in proportion to the age- and gender-specific recommended amount was calculated for each study participant. To evaluate the consumption of sugar-rich foods, the category “total sugars” was constructed by adding all mono- and disaccharides (grams) in the diet, regardless of their source. In the absence of an evaluated reference value for total sugar, the OMD reference value for added sugar was used, corresponding to 6% of the recommended total energy intake [23].

The sex, age group, stage of puberty and migration background distribution of the study population was analyzed. The migration background was classified according to Schenk et al. [31]. Criteria for classification of the parents’ migration background were the following: birthplace, current residence, and first spoken language. To describe the migration background of the study participants themselves, the following criteria were used: German—neither parent with a migration background, one-sided Turkish—one parent with a Turkish migration background, two-sided Turkish - both parents with a Turkish migration background, Other—all other combinations.

Statistical Analysis. In order to investigate the initially constructed dietary pattern throughout the course of the study, the Posteriori Method was chosen for the cluster analysis. Here, the study’s population was divided into individual, non-overlapping clusters based on food consumption. This resulted in every study participant being allocated to a specific cluster [9].

The food groups “total sugar” and “meat/-products” were selected as dietary input variables because it was observed that they possessed the highest interquartile range. The food consumption data of all study participants at the time of enrollment (n = 137) were included in the cluster analysis. The cluster analysis used the calculations generated by SPSS software (IBM SPSS Statistics for Windows, version 25.0, IBM Corp, Armonk, NY, USA). The Ward’s Hierarchical clustering method was applied, which employs Euclidian distances and standard technical settings [32]. First, every unit of observation, in this case the food consumption data adjusted for age and gender of every person in the study, was considered a cluster. Subsequently, the two study patients who were the most similar (Euclidian distance) in their consumption of meat and total sugar were combined into a cluster, so that the variability among the dietary pattern was as minimal as possible. This process was conducted until all study participants were grouped into one cluster. Determining the optimal number of clusters was achieved by observing the decrease in homogeneity within a cluster during the fusion algorithm. For this, the difference in the coefficients of the fusion algorithm during each step of the fusion was calculated and plotted against the cluster number. The optimal cluster number could then be determined from the graphic presentation when a clear bend in the curve—that is, a strong increase in heterogeneity within the new cluster—was observed (Elbow criterion).

To analyze differences in food consumption, weight development (BMI z-score) and insulin resistance (Homeostasis Model Assessment, R-HOMA) over time, mixed-effects models were used which take the correlation of consecutive observations into consideration and use all available data points regardless of any missing values. In these models, dietary pattern, time, treatment group, age, sex and the interaction of time and dietary pattern were used as independent variables (fixed effects). Because of the nonlinear development of the dependent variables, time was used as a categorical variable. Tested contrasts (adjusted means and 95% CI) among dietary pattern regarding food consumption, BMI z-score, and R-HOMA for Δ(T-3,T0), Δ(T0,T12), Δ(T0,T24), and Δ(T12,T24), are reported. For this analysis, SAS PROC GENMOD, SAS Version 9.4 (SAS Institute Inc., Cary, NC, USA) was used. All *p*-values may not be interpreted as confirmative as all analyses were considered exploratory and not adjusted for multiple testing.

## 3. Results

### 3.1. Baseline Charactersistics

The total study population included n = 137 adolescents: 47.4% boys and 52.6% girls. At the time of enrollment (T-3), 38.7% of the participants were 10–12 years of age, 33.6% were 13–14 years of age, and 27.7% were 15–18 years of age. A total of 36.5% of the study participants were affected by obesity (BMI above the 97th percentile) and 63.5% by extreme obesity (BMI above the 99th percentile). The median BMI z-score was 2.52 and the median R-HOMA 2.96. The pubertal status was 11.7% infantile, 37.2% adolescent and 51.1% post-adolescent. A total of 48.9% of the participants were German, 23.4% were two-sided Turkish, 5.1% were one-sided Turkish and 22.7% had some other combination of migration status.

### 3.2. Food Intake at Enrollment, Randomization and Follow-up

Table 1 shows the food consumption of all study participants during the course of the study. At enrollment (T-3), the median intake of the study participants (n = 137) was within a 20% deviation to the food intake recommendations (100%) for the following food groups: vegetables (90%), fruit (87%), bread (89%), milk/dairy products (80%), and butter/margarine/oil (95%). Median intake of starchy foods such as potatoes/pasta/rice (61%) and fish (30%) were considerably under the recommended amounts. Median drinking water consumption (130%) was above the recommendations. Total sugar (411%) as well as meat products intake (173%) far exceeded the recommendations compared to the other food groups. Total sugar and meat products also showed the highest interquartile range, with some participants having consumed very little and others having consumed extremely large amounts of total sugar and/or meat products.

Shortly after the inpatient rehabilitation program (T0), the median consumption of vegetables (154%), fruit (99%), fish (62%), and drinking water (179%) increased in comparison to the time of enrollment (T-3). The median consumption of potatoes/pasta/rice decreased (51%) compared to T-3. The median consumption of meat products (96%) decreased compared to T-3 to a level that corresponded to the recommendations. The consumption of total sugar (298%) decreased compared to T-3 to a level three times higher than the recommendations.

12 months after the inpatient rehabilitation program (T12), the median intake of vegetables (92%) and drinking water (133%) decreased compared to T0. The median consumption of total sugar (322%) and meat products (148%) increased again compared to T0, however, not to the consumption level at the starting point of T-3. In the follow-up 24 months after the inpatient rehabilitation program (T24), the median consumption of total sugar (261%) decreased compared to T12, while the median consumption of meat products (190%) increased compared to T12. Fruits (61% and 49%) were consumed significantly less at T12 as well as at T24 compared to enrollment (T-3) so that the consumption level no longer met the recommendations. Similarly, the median consumption of potatoes/pasta/rice (49% and 50%), bread and cereals (69% and 54%), milk/dairy products (73% and 72%), fish (59% and 24%), as well as butter/margarine/oil (73% and 71%) in both follow-ups (T12 and T24) was below the recommendations.

There were no significant differences in food consumption between the intervention and control group at enrollment and after 12 months of intervention (results not shown).

### 3.3. Initial Dietary Pattern

No differences in food consumption were observed between the intervention and the control group from T0 to T12. For this reason, the food consumption data from all participants at the time T-3 were used in the cluster analysis and were not differentiated by the intervention and control group.

In order to cluster different dietary patterns, the food groups meat/-products and total sugar were included in the cluster analysis, which showed the highest interquartile range regarding food consumption within the study population. The cluster analysis according to Ward’s method yielded a 3-cluster solution (Figure 2).

There were no differences between the dietary patterns in the propensities of sex, age group, stage of puberty and migration background. The naming of the clusters was done following Ruiz-Tovar et al. [15]. The highest food consumption and consequently the highest total energy intake was observed in cluster 1 (n = 32). Therefore, the cluster has been named Big Eaters.

Big Eaters consumed significantly more potatoes/pasta/rice, milk/dairy products, meat/-products, butter/margarine/oil as well as total sugar than cluster 2 (n = 66) and cluster 3 (n = 39). Cluster 2 had the lowest total calorie intake of all clusters and has been named Moderate Eaters. Cluster 3 showed the lowest consumption of meat/-products and ranged between the Big Eaters and the Moderate Eaters regarding the consumption of total sugar. The high total sugar was the prominent characteristic of this cluster, which has therefore been named Snackers.

Table 2 shows the consumption of sweets (in g/d) and sugar-sweetened beverages (in mL/d). The Big Eaters showed the highest consumption of sweets and sugar-sweetened beverages at enrollment, followed by the Snackers, while the lowest consumption of these foods was observed in Moderate Eaters.

### 3.4. Changes in Food Consumption, BMI z-Score and R-HOMA among Clusters

As shown in Figure 3a, after inpatient rehabilitation, Big Eaters and Snackers had significantly reduced their consumption of total sugar. Total sugar consumption after rehabilitation (T0) was nearly the same in all dietary patterns with an amount that was 3 times higher than recommended. Between T12 an T24 in all dietary patterns the intake of total sugar decreased significantly. The consumption of meat products (Figure 3b) also decreased significantly during inpatient rehabilitation for all dietary patterns. However, it increased again significantly up to T12.

The BMI z-score (Figure 3c) decreased significantly in all three dietary patterns from T-3 to T0 (*p* < 0.0001) and increased again significantly up to T12. From T12 to T24, the BMI z-score tended to increase in Big Eaters (*p* = 0.068). No differences in R-HOMA between the individual clusters was observed at the different points in time of the study. After inpatient rehabilitation, R-HOMA was reduced for all dietary patterns and increased again after one year (Figure 3d).

## 4. Discussion

This research started based on the following assumptions: different dietary patterns exist within a group of adolescents with obesity, and with regard to the changes in diet, each dietary pattern benefits to different degrees from the inpatient weight loss program.

Three distinct dietary patterns in adolescents with obesity were identified using cluster analysis: Big Eater, Snacker, and Moderate Eater. In all dietary patterns, the consumption of meat/-products and total sugars exceeded the recommendation for an optimized balanced diet [23]. On a national average, adolescents in Germany eat over twice the recommended amount with respect to these two food groups [26]. Among the Big Eaters and Snackers, consumption of meat/-products reduced in the medium term after inpatient rehabilitation, as did consumption of high-sugar foods in the long term. In the Moderate Eater group, whose food consumption of meat and sweets is comparable to the national average in Germany, no improvement in food consumption was observed after inpatient rehabilitation.

Big Eaters and Snackers consumed significantly too much total sugar before the inpatient rehabilitation. Adolescents in both dietary patterns were able to reduce sugar consumption in the long term, with the greatest changes observed in Big Eaters with regard to the reduction of sugar-sweetened beverages. The Big Eaters initially consumed a median intake of approximately 1000 mL of sweetened beverages per day, which is roughly equivalent to 100 g of sugar and 400 kilocalories [33]. Significant reductions in sugar-sweetened beverages led to the conservation of long-term calories and lowers the risk of obesity-associated comorbidities [34]. As shown in the present work, both the reduction of total sugars and weight reduction improved the R-HOMA index as a measure of insulin sensitivity in all dietary patterns. The results lead to the question of why the adolescents succeeded in reducing high-sugar foods over the long term. One hypothesis is that the significant reduction in sugar during the inpatient rehabilitation caused a change in taste perception and that this change in perception helped to transfer new learned behaviours into daily life, such as drinking water instead of sugar-sweetened beverages and eating smaller amounts of sweets. This assumption is supported in a study by Kalveram et al. [35], which showed that the perception of sweet taste improves with a stay in inpatient rehabilitation. There is potential here for further studies to investigate the change process in nutrition using qualitative research methods and to identify effective factors that contribute to a sustainable reduction in sugar consumption.

It should be noted that the DISHES software does not allow for the distinction between “free sugar” or “added sugar” in foods. Therefore, in the present work, all mono- and disaccharides were added together, regardless of their source, and declared as “total sugar”. Since there is no uniform recommendation for “total sugar”, the recommendation of 6% “added sugar” of the total recommended energy was used instead [23,36]. “Added sugar” includes all sugars that have been additionally added during the processing of foods and excludes naturally occurring sugars in fruits, vegetables, and dairy products [37]. We have chosen total sugar because this corresponds to the advice given in the outpatient obesity clinic. Here, families are trained to pay attention to the total sugar content— for example, a glass of fruit juice corresponds to the sugar that is allowed for the day. In this way, it can also be explained that directly after inpatient rehabilitation, the amount of total sugar consumed by the study participants is above the reference for added sugar.

In contrast to the long-term reduced sugar consumption, the consumption of meat/-products increased again after the inpatient rehabilitation. Thus, the adolescents were unsuccessful in maintaining a reduced consumption of meat/-products after the inpatient rehabilitation. One possibility is that meat consumption increases after inpatient rehabilitation because adolescents have returned to eating fast food more often. Adolescents’ health literacy is also influenced by their parents [38]. It can be hypothesized that adolescents of parents with obesity are used to larger serving sizes during family meals [39] As a result, it is likely that they eat too much meat in total. With this in mind, parents should be educated early on about age-appropriate portion sizes by nutritionists [39,40]. It would be also worthwhile to investigate the extent to which training parents on high-protein alternatives has a long-term effect on reducing meat consumption. 

The Moderate Eater dietary pattern showed no further improvement in terms of their diet after the inpatient rehabilitation. It is possible that a significant portion of adolescents assigned to the Moderate Eater dietary pattern underreport their dietary intake. Studies suggest that underreporting is higher among adolescents who are affected by overweight or obesity than among those who are classified as normal weight or underweight [41,42]. This could explain why the largest portion of adolescents with obesity was assigned to the Moderate Eater dietary pattern. Further research might investigate how impactful a focus on regular physical activity and exercise for Moderate Eaters can be to supplement the basic dietary recommendations. In this context, it would be valuable to examine the extent to which Moderate Eaters differ from adolescents of the national average in terms of their regular physical activity.

Another potential explanation to the Moderate Eater dietary pattern having shown no improvement in terms of their diet after the inpatient rehabilitation is the following. In contrast to the Big Eaters and Snackers, the Moderate Eaters had a diet at the time of recruitment that was comparable to the national average in Germany [26]. With their diet comparable to the norm, it is conceivable that the Moderate Eaters were less able to realize a functional and sustainable dietary strategy after the inpatient rehabilitation compared with the Big Eaters and Snackers. While all dietary patterns received basic dietary recommendations from the individualized nutrition therapy in the outpatient setting, these recommendations were further away from the eating habits of the Big Eaters and Snackers. Therefore, it might have been easier for the Big Eaters and Snackers to leave the inpatient rehabilitation with clearer and more actionable takeaways. For example, the Big Eaters might have more easily realized recognizing appropriate portion size and regularly drinking water instead of sugar sweetened beverages. The Snackers might have benefited from learning to have more regular meals and eating only small amounts of sweets. For Moderate Eaters, the takeaways might have been less obvious since their eating habits were closer to the national average and therefore the dietary recommendations. Since the takeaways were less obvious, they were more challenging to actualize for this dietary pattern.

Strengths and Limitations. The majority of studies examining dietary patterns are cross-sectional studies [14]. In this study, the alteration in food consumption from initial dietary patterns was observed over a longer period of time. The findings about which alterations in dietary patterns are not sustainable can be transferred to the outpatient setting and followed up with families. The nutrition interview was conducted face-to-face by experienced nutritionists, using model dishes to determine portion sizes. The food consumption data is more detailed than that of an FFQ, which is commonly used in similar studies. However, there are some limitations. The diet history interviews were conducted by experienced dietitians, but misreporting may still occur due to memory lapses and socially desirable responses. Underreporting during the course of the study can also not be ruled out due to the learning effect of the study participants. DISHES is validated for adults only [27], although estimated food consumption showed fairly good agreement with results based on an FFQ among adolescents also [28]. The influence of physical activity, which has a major impact on energy balance and weight gain after the inpatient rehabilitation, was not investigated in this work. 

## 5. Conclusions

The consumption of meat and sweets is not only too high among adolescents with obesity, but also exceeds twice the recommendation in a national comparison of adolescents in Germany [26]. These two food groups should therefore play an important role in outpatient obesity therapy. As the results show, meat could be replaced by dairy products, fish or legumes, since the overall consumption of these foods is too low. Since adolescents do not have an independent choice of food at home, parents need to be trained to choose healthier foods and implement health-oriented food rules [43]. It would be worthwhile to investigate the impact that training parents in an outpatient setting on high-protein alternatives to meat can have on the long-term effect of reducing meat consumption during family meals. Parents also under-estimate the sugar content of foods [44]. In outpatient nutritional counselling, families and adolescents should be trained to investigate the sugar content of frequently consumed foods and to explore alternative food products containing less sugar. Compared with the Big Eater and Snacker groups, the Moderate Eaters showed no further improvements in the long term with respect to their diet. However, Moderate Eaters were above the recommendation with their consumption of meat/-products and sweets to a similar extent as the national average in Germany [26]. From clinical experience, keeping the patient’s motivation high is key to success. Since the Moderate Eater group has a diet comparable to the national average in Germany, further dietary restrictions for this group could lead to frustration and a risk of regressing. For the Moderate Eaters, a shift in focus towards increased physical activity is potentially a more important starting point in outpatient obesity therapy for weight stability or further weight loss.

## Figures and Tables

**Figure 1 ijerph-19-16613-f001:**
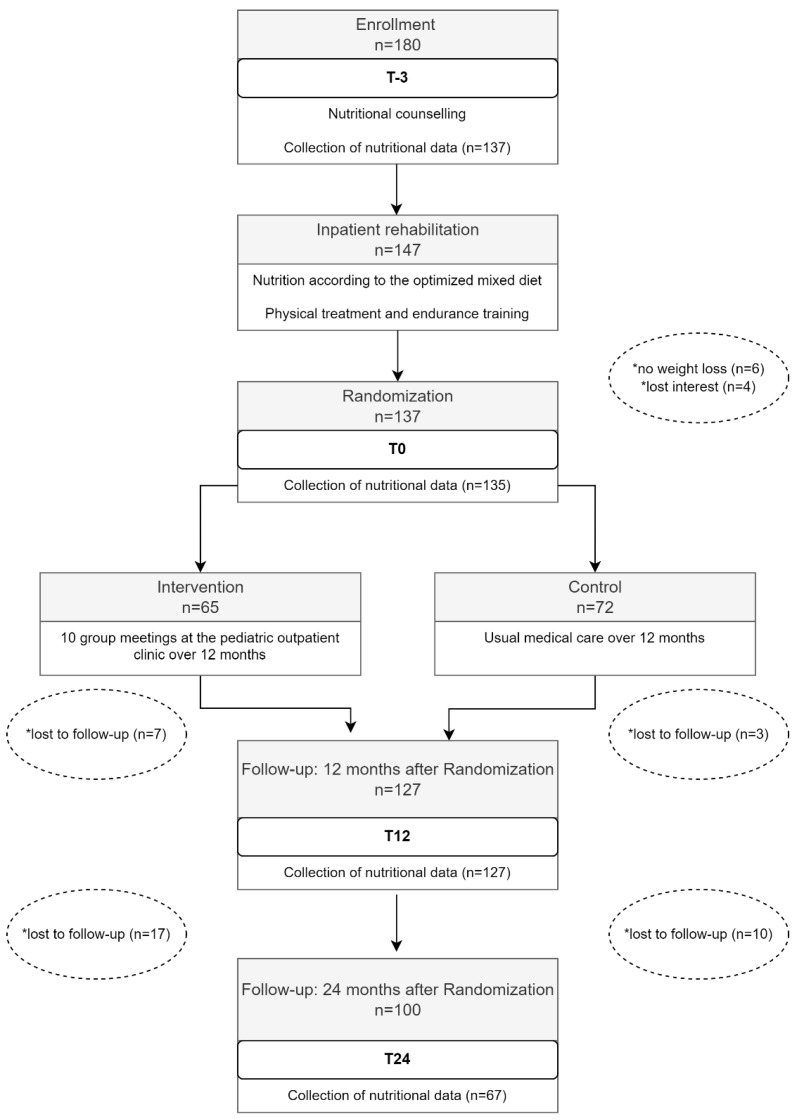
Collection of dietary intake data within the framework of the Maintain study. * Reasons for drop-out of study participants.

**Figure 2 ijerph-19-16613-f002:**
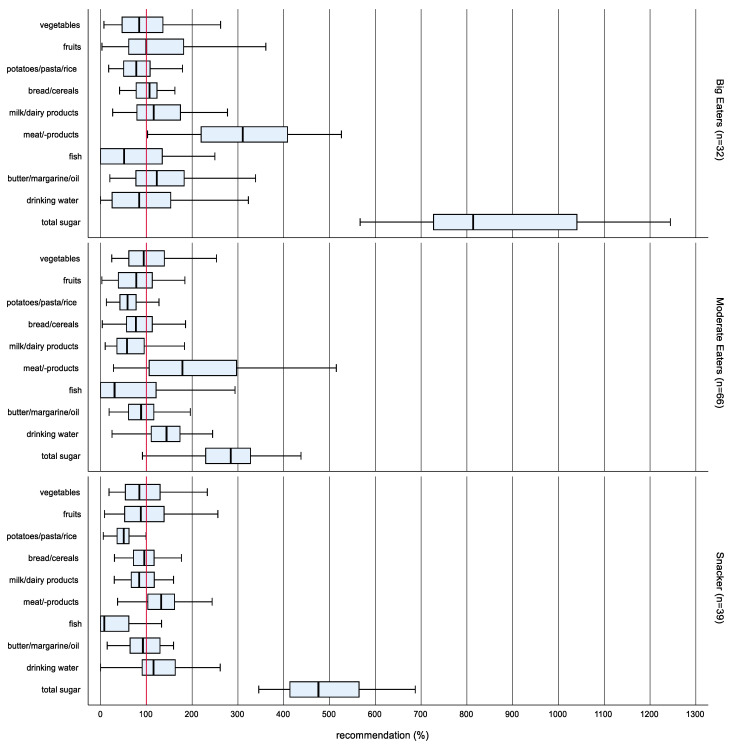
Box-Whisker plots * of the intake in proportion to the OMD recommended amount according to dietary patterns at enrollment (n = 137). * Box-Whisker plots with the inner box corresponding to the interquartile range around the median and the whisker corresponding to the 5th and 95th percentiles (pct).

**Figure 3 ijerph-19-16613-f003:**
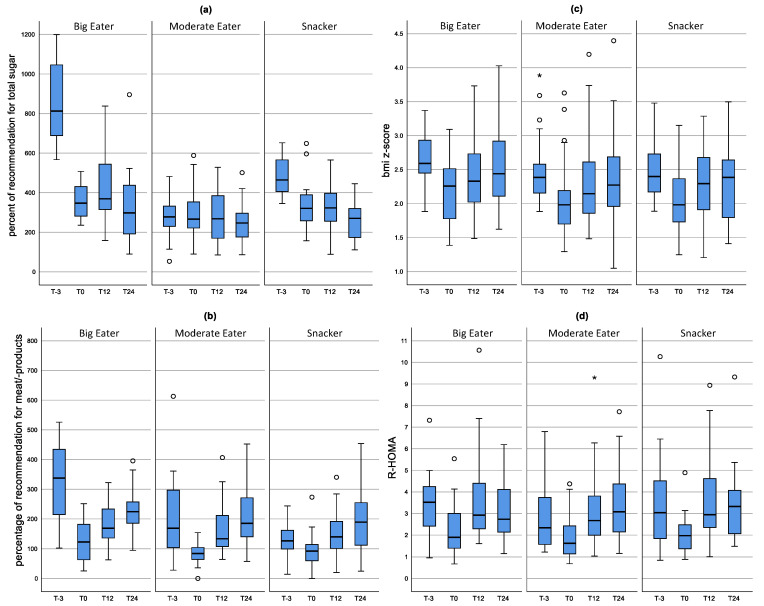
Distribution of consumption of meat products (**a**) and total sugar (**b**) as well as the BMI z-score (**c**) and R-HOMA (**d**) during the course of the study according to dietary patterns. Results of the mixed model. The mean value as well as the upper and lower limit are shown. Box-Whisker plots with the inner box corresponding to the interquartile range around the median and the whiskers corresponding to the 5th and 95th percentiles.

**Table 1 ijerph-19-16613-t001:** Food consumption of the study population at enrollment (T-3), directly after intensive inpatient rehabilitation (T0) and at 12-month (T12) as well as 24 months follow up (T24).

	T-3 (n = 137)			T0 (n = 135)			T12 (n = 127)			T24 (n = 67)		
Food Group	Q0.25	Q0.5	Q0.75	Q0.25	Q0.5	Q0.75	Q0.25	Q0.5	Q0.75	Q0.25	Q0.5	Q0.75
veggie	57.5	90.0	137.1	106.3	154.4	194.2	54.2	92.2	131.4	56.9	94.1	150.3
fruit	50.0	86.6	129.7	59.6	98.9	137.0	37.2	61.1	100.5	28.6	49.4	76.2
starch	41.6	61.1	83.0	41.5	51.3	60.1	35.8	48.7	64.2	33.5	50.4	61.8
grain	65.7	89.3	117.2	67.8	77.8	95.3	50.4	68.6	96.6	41.9	53.9	71.7
milk	48.6	80.1	116.5	53.7	76.0	102.5	48.5	72.6	102.8	49.1	72.0	94.5
meat	114.3	173.0	297.1	63.8	95.8	133.6	107.1	148.4	228.9	126.2	190.9	253.0
fish	0.0	29.7	113.6	22.5	61.6	101.8	0.0	59.3	103.2	0.0	24.3	89.9
fat	67.9	94.8	135.2	46.6	71.9	99.5	48.3	73.1	105.4	40.9	71.4	117.1
water	88.7	129.5	166.6	141.7	178.6	212.5	89.3	133.3	175.9	101.3	129.2	171.9
sugar	287.2	411.2	610.2	237.1	297.9	388.5	205.6	321.5	430.1	185.8	261.3	333.1

Note: Data are reported as 25th quartile (Q0.25), median (Q0.5), and 75th quantile (Q0.75) for the food groups vegetables (veggie), fruits (fruit), potatoes, pasta and rice (starch), bread and cereals (grain), milk and milk products (milk), meat and cold meats (meat), fish (fish), butter, margarine and oil (fat), drinking water (water), and total sugar (sugar).

**Table 2 ijerph-19-16613-t002:** Consumption of sweets and sugar-sweetened beverages (SSB) of the dietary patterns at enrollment (T-3), directly after inpatient rehabilitation (T0), and at follow-up (T12, T24).

	Big Eater			Moderate Eater			Snacker		
Sweets [g/d]	Q02.5	Q0.5	Q0.75	Q02.5	Q0.5	Q0.75	Q02.5	Q0.5	Q0.75
T-3	75.6	97.7	165.9	18.8	35.9	62.7	38.6	67.8	104.2
T0	14.7	33.0	51.0	12.4	24.5	46.1	9.5	24.7	53.5
T12	28.9	47.2	90.8	27.4	48.1	70.9	19.7	43.6	86.6
T24	27.1	43.7	76.7	27.7	36.1	60.6	18.2	41.8	75.7
**SSB [g/d]**	**Q02.5**	**Q0.5**	**Q0.75**	**Q02.5**	**Q0.5**	**Q0.75**	**Q02.5**	**Q0.5**	**Q0.75**
T-3	555.0	1042.4	1906.3	1.2	72.5	294.5	85.9	238.8	524.5
T0	0.9	23.8	44.2	0.8	1.5	19.3	1.5	17.7	36.8
T12	48.9	211.7	465.4	1.5	127.8	270.0	0.9	161.7	357.8
T24	19.6	163.6	348.7	0.4	24.9	215.0	47.3	124.4	358.7

Note: Data are reported as 25th quartile (Q02.5), median (Q0.5) and 75th quantile (Q0.75).

## Data Availability

The datasets generated during and/or analyzed during the currentwork are not publicly available but are available from the corresponding author on reasonable request.

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
