# Peer review of "Dietary Patterns in Adolescent Obesity as Predictors of Long-Term Success Following an Intensive Inpatient Lifestyle Programme"

_ijerph, 2022, doi:10.3390/ijerph192416613_

Round 1
Reviewer 1 Report
Dear Authors,
The study in my opinion is very interesting. There are some very important elements that I think should be improved.
Please present Table 1 and Table 2 in a more readable way. Please use separate columns and not division marks.
The discussion should include the weaknesses and strengths of the study.
The conclusions should be more precise. Please add a practical conclusion of the use of the study.
The literature used in the publication is old. It is worth reviewing new publications in this field and enriching the study with them. At least 75% should be from the last 10.
Author Response
Thank you for the constructive feedback that helped us to revise and improve the paper. We hope that the study insights are now more easily understandable to the reader.
The study in my opinion is very interesting. There are some very important elements that I think should be improved.
Please present Table 1 and Table 2 in a more readable way. Please use separate columns and not division marks.
Table 1 and Table 2 have been revised as recommended in a more readable way.
The discussion should include the weaknesses and strengths of the study.
We added the following paragraph about strengths and limitations at the end of the discussion:
Strengths and Limitations. The majority of studies examining dietary patterns are cross-sectional (Quelle). In this experimental study, different dietary patterns were observed over an extended period of time. The nutrition interview was conducted face-to-face by experienced nutritionists using tableware models to determine portion sizes. The food consumption data is more detailed than that of an FFQ, which is commonly used in similar studies. However, there are some limitations. While the diet history interviews were conducted by experienced dietitians, misreporting may still occur due to memory lapses and socially desirable responses. Underreporting during the course of the study can also not be ruled out due to the learning effect of the study participants. DISHES is validated for adults only (37), although estimated food consumption also showed fairly good agreement with results based on an FFQ among adolescents (38). The influence of physical activity, which has a major impact on energy balance and weight gain after the inpatient rehabilitation, was not investigated in this work.
The conclusions should be more precise. Please add a practical conclusion of the use of the study.
The following paragraph was added to the conclusion:
The consumption of meat and sweets is not only too high among adolescents with obesity, but also exceeds twice the recommendation in a national comparison of adolescents in Germany [25, 40]. These two food groups should therefore play an important role in outpatient obesity therapy. As the results show, meat could be replaced by dairy products, fish or legumes, since the overall consumption of these foods is too low. Since adolescents do not have an independent choice of food at home, parents need to be trained to choose healthier foods and implement health-oriented food rules [41]. It would be worth-while to investigate the impact that training parents in an outpatient setting on high-protein alternatives to meat can have on the long-term effect of reducing meat consumption during family meals. Parents also under-estimate the sugar content of foods [42]. In outpatient nutritional counselling, families and adolescents should be trained to investigate the sugar content of frequently consumed foods and to explore alternative food products containing less sugar. Compared with the Big Eater and Snacker groups, the Moderate Eaters showed no further improvements in the long term with respect to their diet. However, Moderate Eaters were above the recommendation with their consumption of meat/-products and sweets to a similar extent as the national average in Germany [25]. From clinical experience, keeping the patient’s motivation high is key to success. Since the Moderate Eater group has a diet comparable to the national average in Germany, further dietary restrictions for this group could lead to frustration and a risk of regressing. For the Moderate Eaters, a shift in focus towards increased physical activity is potentially a more important starting point in outpatient obesity therapy for weight stability or further weight loss.
The literature used in the publication is old. It is worth reviewing new publications in this field and enriching the study with them. At least 75% should be from the last 10.
Thank you for the important comment. We updated the literature research and replaced older references with more current literature.
We have left some necessary, old references pointing to the original literature in the methods section in the article. These include, for example, the determination of pubertal status according to Tanner et al. and the German reference population according to Kromeyer et al. for the determination of BMI percentiles.
Reviewer 2 Report
Overall this is a well written paper reviewing dietary patterns of adolescents with obesity and how this may predict longer term changes after an inpatient intervention.
There are a few instances in the paper where the language needs to be changed to person first language eg in the introduction paragraph lines 37/38 - instead of "remain obese" should be changed to "continue to live with obesity" or a similar phrase. Similarly line 52 change from "the risk of becoming overweight or obese" to "risk of developing overweight or obesity".
In the materials and methods section why is "randomized controlled trial" in quotation marks? This implies there was something irregular and not a true RCT? Can the authors explain further or otherwise remove the quotation marks.
Also in the methods section it would be helpful to include the inclusion criteria to give readers and easy reference as to the cohort composition.
Line 87 the kg in relation to the stadiometer measurement should be changed to cm.
In figure 1 the last box indicates collection of data in 67 participants however in the text says 66. Is this just a typo?
The authors describe the participants having 5-6 regular meals however what they really mean is 3 meals and in-between snacks. Using the word meals for snacks suggests more substantive eating.Would be more correct to either describe as above or refer to 5-6 regular eating episodes.
The sentence referring to DISHES (enables to assess the individual the average food consumption......) does not make sense. Presumption an english translation glitch?
I find it hard to believe that the adolescent's intake of vegetables was 90% of the recommended intake given data from many western countries indicate a lower intake level (either a lower percentage of adolescents eating vegetables or those that do eat vegetables consume a much smaller percentage of the recommended intake). Is this in keeping with the national average for Germany? I also don't agree with statements that the moderate eaters have a diet comparable to the national average (as stated in the abstract and the discussion in particular) however reviewing the data in figure 2 the consumption of meat products is still at 200% median and total sugar is at almost 300% median, which are the two main products that are being focused on in the paper. And thus this group still have an exceedingly intake compared with the average population. In view of this I would expect the discussion to reflect this rather than almost ignore it and suggest that a focus on physical activity is needed in this group. They still require a reduction in those two key food groups as do the other two cohorts. More discussion around why they're eating habits did not change would be interesting?
How generalisable are these findings - not all weight management intervention services have the luxury of an inpatient program. How do the authors findings translate to the outpatient sector? Given the follow up was over 24 months, (which is excellent data to have) does this make the results potentially more generalisable for the outpatient intervention setting?
I believe addressing the comments above, particularly in relation to the inferences in relation to the moderate eaters, would improve this paper.
Author Response
Thank you for the constructive feedback that helped us to revise and improve the paper. We hope that the study insights are now more easily understandable to the reader.
Overall this is a well written paper reviewing dietary patterns of adolescents with obesity and how this may predict longer term changes after an inpatient intervention.
There are a few instances in the paper where the language needs to be changed to person first language eg in the introduction paragraph lines 37/38 - instead of "remain obese" should be changed to "continue to live with obesity" or a similar phrase. Similarly line 52 change from "the risk of becoming overweight or obese" to "risk of developing overweight or obesity".
We have reviewed the paper with regard to the person first language and changed the following wording as recommended:
- "remain obese" was changed to "continue to live with obesity"
- "the risk of becoming overweight or obese" was changed to "risk of developing overweight or obesity"
In the materials and methods section why is "randomized controlled trial" in quotation marks? This implies there was something irregular and not a true RCT? Can the authors explain further or otherwise remove the quotation marks.
Thank you for the observation. The quotation marks should not have been there and have been removed.
Also in the methods section it would be helpful to include the inclusion criteria to give readers and easy reference as to the cohort composition.
We made a reference to the study design paper for further information and added the inclusion criteria for the study as follows:
Of these, 147 adolescents and their families met the following inclusion criteria: Age between 10 to 17 years, BMI above the 97th percentile at recruitment, active participation in the inpatient rehabilitation program as well as the intervention and control group and free living part.
Line 87 the kg in relation to the stadiometer measurement should be changed to cm.
We have modified the paragraph as follows:
Anthropometric data of the participants were measured in a standardized way. Body weight was measured while participants were wearing light underwear, without shoes on a digital scale (Soehnle, Germany) with an accuracy of 0.1 kg. Body height was measured with a wall-mounted stadiometer (Keller, Germany) with an accuracy of 0.1 cm.
In figure 1 the last box indicates collection of data in 67 participants however in the text says 66. Is this just a typo?
The number in the figure is correct. We have changed the sentence, as follows, to explain how the number of DISHES interviews came about:
Until the follow-up at time T24, another 27 study participants lost interest and dropped out leaving 100 study participants remaining. 79 DISHES interviews were conducted from the remaining 100 study participants. Only 67 of the 79 DISHES interviews could be considered for this study at T24, due to 12 of the 79 participants having attended inpatient rehabilitation twice since time point T12.
The authors describe the participants having 5-6 regular meals however what they really mean is 3 meals and in-between snacks. Using the word meals for snacks suggests more substantive eating.Would be more correct to either describe as above or refer to 5-6 regular eating episodes.
Thanks for this important wording note. We changed the paragraphs as follows:
Three main meals were offered daily, consisting of a cooked lunch and one bread meal for breakfast as well as dinner. Between the main meals, two fruit and one tea snack were offered.
The sentence referring to DISHES (enables to assess the individual the average food consumption......) does not make sense. Presumption an english translation glitch?
Yes, it was a translation glitch. We corrected the sentence as follows:
DISHES enables an assessment of the participant’s average food consumption over a four week period within a computer based diet history interview.
I find it hard to believe that the adolescent's intake of vegetables was 90% of the recommended intake given data from many western countries indicate a lower intake level (either a lower percentage of adolescents eating vegetables or those that do eat vegetables consume a much smaller percentage of the recommended intake). Is this in keeping with the national average for Germany?
Our group of adolescents with obesity showed a higher median consumption of vegetables and fruits than the national average for Germany, which was 90% of the recommendation. We were also surprised by this.
Some possible explanations for this result:
- Many of the adolescents were recruited at the Charité obesity outpatient clinic and often had prior counselling on nutrition. Accordingly, the families had already tried to increase the proportion of vegetables beforehand.
- A relatively high proportion of adolescents have a migration background. From practical experience in nutrition counselling, we know that the main meal is often cooked with an abundance of vegetables and served with a salad in families with a Turkish and Arabic migration background.
- Big Eaters consume larger portion sizes. This leads to a higher total consumption of vegetables, despite the proportion of vegetables compared to their total intake still being low.
I also don't agree with statements that the moderate eaters have a diet comparable to the national average (as stated in the abstract and the discussion in particular) however reviewing the data in figure 2 the consumption of meat products is still at 200% median and total sugar is at almost 300% median, which are the two main products that are being focused on in the paper. And thus this group still have an exceedingly intake compared with the average population. In view of this I would expect the discussion to reflect this rather than almost ignore it and suggest that a focus on physical activity is needed in this group. They still require a reduction in those two key food groups as do the other two cohorts. More discussion around why they're eating habits did not change would be interesting?
On a national average in Germany, adolescents' consumption of meat is almost 200% of the recommendation and high-sugar and high-fat snacks is about 250% of the recommendation. Regarding the consumption of basic foods such as dairy products, fruits, vegetables, and carbohydrate-rich foods, the national average in Germany does not reach the recommendations.
Based on the results of Mensink et al., the diet of the moderate eaters group was described in the paper as comparable to the national average.
We’ve made comparison between the food consumption of Moderate Eaters and the national average more clear in this section of the discussion as follows:
On a national average, adolescents in Germany also eat over twice the recommended amount with respect to these two food groups. Among the Big Eaters and Snackers, consumption of meat/-products reduced in the medium term after inpatient rehabilitation, as did consumption of high-sugar foods in the long term. The Moderate Eaters, whose food consumption of meat and sweets is comparable to the national average in Germany, showed no improvement in food consumption after inpatient rehabilitation.
We also added further explanations to the discussion regarding the lack of dietary improvement by the Moderate Eaters compared with the other dietary patterns, as well as ideas for further follow-up research:
The Moderate Eater dietary pattern showed no further improvement in terms of their diet after the inpatient rehabilitation. It is possible that a significant portion of adolescents assigned to the Moderate Eater dietary pattern underreport their dietary intake. Studies suggest that underreporting is higher among adolescents who are affected by overweight or obesity than among those who are classified as normal weight or underweight [40, 41]. This could explain why the largest portion of adolescents with obesity was assigned to the Moderate Eater dietary pattern. Further research might investigate how impactful a focus on regular physical activity and exercise for Moderate Eaters can be to supplement the basic dietary recommendations. In this context, it would be valuable to examine the extent to which Moderate Eaters differ from adolescents of the national average in terms of their regular physical activity.
Another potential explanation to the Moderate Eater dietary pattern having shown no improvement in terms of their diet after the inpatient rehabilitation is the following. In contrast to the Big Eaters and Snackers, the Moderate Eaters had a diet at the time of recruitment that was comparable to the national average in Germany [25]. With their diet comparable to the norm, it is conceivable that the Moderate Eaters were less able to realize a functional and sustainable dietary strategy after the inpatient rehabilitation compared with the Big Eaters and Snackers. While all dietary patterns received basic dietary recom-mendations from the individualized nutrition therapy in the outpatient setting, these recommendations were further away from the eating habits of the Big Eaters and Snackers. Therefore, it might have been easier for the Big Eaters and Snackers to leave the inpatient rehabilitation with clearer and more actionable takeaways. For example, the Big Eaters might have more easily realized recognizing appropriate portion size and regularly drinking water instead of sugar sweetened beverages. The Snackers might have benefited from learning to have more regular meals and eating only small amounts of sweets. For Moderate Eaters, the takeaways might have been less obvious since their eating habits were closer to the national average and therefore the dietary recommendations. Since the takeaways were less obvious, they were more challenging to actualize for this dietary pattern.
How generalisable are these findings - not all weight management intervention services have the luxury of an inpatient program. How do the authors findings translate to the outpatient sector? Given the follow up was over 24 months, (which is excellent data to have) does this make the results potentially more generalisable for the outpatient intervention setting?
I believe addressing the comments above, particularly in relation to the inferences in relation to the moderate eaters, would improve this paper.
We added a practical conclusion of the study as follows:
The consumption of meat and sweets is not only too high among adolescents with obesity, but also exceeds twice the recommendation in a national comparison of adolescents in Germany [25, 40]. These two food groups should therefore play an important role in outpatient obesity therapy. As the results show, meat could be replaced by dairy products, fish or legumes, since the overall consumption of these foods is too low. Since adolescents do not have an independent choice of food at home, parents need to be trained to choose healthier foods and implement health-oriented food rules [41]. It would be worth-while to investigate the impact that training parents in an outpatient setting on high-protein alternatives to meat can have on the long-term effect of reducing meat consumption during family meals. Parents also under-estimate the sugar content of foods [42]. In outpatient nutritional counselling, families and adolescents should be trained to investigate the sugar content of frequently consumed foods and to explore alternative food products containing less sugar. Compared with the Big Eater and Snacker groups, the Moderate Eaters showed no further improvements in the long term with respect to their diet. However, Moderate Eaters were above the recommendation with their consumption of meat/-products and sweets to a similar extent as the national average in Germany [25]. From clinical experience, keeping the patient’s motivation high is key to success. Since the Moderate Eater group has a diet comparable to the national average in Germany, further dietary restrictions for this group could lead to frustration and a risk of regressing. For the Moderate Eaters, a shift in focus towards increased physical activity is potentially a more important starting point in outpatient obesity therapy for weight stability or further weight loss.